

# Aerosol measurement methods to quantify spore emissions from fungi and cryptogamic covers in the Amazon

Nina Löbs[1*], Cybelli G. G. Barbosa[1*], Sebastian Brill[1], David Walter[1], Florian Ditas[1], Marta de Oliveira Sá[2], Alessandro C. de Araújo[3], Leonardo R. de Oliveira[2], Ricardo H. M. Godoi[4], Stefan Wolff[1], Meike Piepenbring[5], Jürgen Kesselmeier[1], Paulo Artaxo[6], Meinrat O. Andreae[1,7], Ulrich Pöschl[1], Christopher Pöhlker[1], Bettina Weber[1,8*]

1 Multiphase Chemistry and Biogeochemistry Departments, Max Planck Institute for Chemistry, Mainz, 55128, Germany

2 Large Scale Biosphere-Atmosphere Experiment in Amazonia (LBA), Instituto Nacional de Pesquisas da Amazonia (INPA), Manaus-AM, CEP 69067-375, Brazil

3 Empresa Brasileira de Pesquisa Agropecuária (EMBRAPA), Belém-PA, CEP 66095-100, Brazil

4 Department of Environmental Engineering, Federal University of Paraná UFPR, Curitiba, PR, Brazil

5 Department of Mycology, Goethe University Frankfurt/Main, Frankfurt, 60438, Germany

6 Institute of physics, University of São Paulo 05508-900, Brazil

7 Scripps Institution of Oceanography, University of California San Diego, La Jolla, CA 92037, US

8 Institute of Plant Sciences, University of Graz, Holteigasse 6, 8010 Graz, Austria

*Correspondence to*: Nina Löbs (n.loebs@mpic.de), Christopher Pöhlker (c.pohlker@mpic.de), Bettina Weber (b.weber@mpic.de)



**Abstract.** Bioaerosols are considered to play a relevant role in atmospheric processes, but their sources, properties and spatio-temporal distribution in the atmosphere are not yet well characterized. In the Amazon Basin, primary biological aerosol particles (PBAP) account for a large fraction of coarse particulate matter, and fungal spores are among the most abundant PBAP there as well as in other vegetated continental regions. Furthermore, PBAP could also be important ice nuclei in Amazonia. Measurement data on the release of fungal spores under natural conditions, however, are sparse. Here we present an experimental approach to analyze and quantify the spore release from fungi and other spore producing organisms under natural and laboratory conditions. For measurements under natural conditions, the samples were kept in their natural environment and a setup was developed to estimate the spore release numbers and sizes together with the microclimatic factors temperature and air humidity, as well as the mesoclimatic parameters net radiation, rain, and fog occurrence. For experiments in the laboratory, we developed a cuvette to assess the particle size and number of newly released fungal spores under controlled conditions, simultaneously measuring temperature and relative humidity inside the cuvette. Both approaches were combined with bioaerosol sampling techniques to characterize the released particles by microscopic methods. For fruiting bodies of the basidiomycetous species, *Rigidoporus microporus*, the model species for which these techniques were tested, the highest frequency of spore release occurred in the range of 62 and 96 % relative humidity. The results obtained for this model species reveal characteristic spore release patterns linked to environmental or experimental conditions, indicating that the moisture status of the sample may be a regulating factor, while temperature and light seem to play a minor role for this species. The presented approach enables systematic studies aimed at the quantification and validation of spore emission rates and inventories, which can be applied to a regional mapping of cryptogamic organisms under given environmental conditions.

## 1    Introduction

Particles released from the biosphere into the atmosphere are called primary biological aerosol particles (PBAP). They comprise biogenic matter such as pollen, bacteria, viruses, spores of fungi, bryophytes, and ferns, cells and cell compounds (e.g., proteins) of plants and animals (Andreae and Crutzen, 1997; Després et al., 2012; Fröhlich-Nowoisky et al., 2016; Simoneit, 1977, 1989). They can influence the water cycle by acting as cloud condensation or ice nuclei, thereby triggering precipitation (Möhler et al., 2007; Pöschl et al., 2010) and they can affect the radiative budget of the atmosphere, by the absorption and scattering of radiation, on a local and global scale (Cox, 1995).

Although the general knowledge on PBAP is still sparse, there are several studies on the sampling and sizing of PBAP in the Amazon (Elbert et al., 2007; Gilbert and Reynolds, 2005; Graham et al., 2003; Huffman et al., 2012b; Moran-Zuloaga et al., 2018; Pöschl et al., 2010; Whitehead et al., 2016; Womack, 2015). However, a large variety of techniques has been utilized in these aerosol sampling studies, which handicaps their direct comparability. For online counting and/or sizing of aerosols at different heights within and above of the canopy, the sensitive analyzers are mostly situated in weather-protected containers at the forest ground. The required long inlet lines from a tower or mast cause a preferential loss of larger particles (Huffman et al., 2012b; Moran-Zuloaga et al., 2018; Pöschl et al., 2010). Other aerosol collection devices (e.g. impinger samplers) can be located at the desired sampling height without extended inlet lines (Elbert et al., 2007; Gilbert and Reynolds, 2005; Graham





et al., 2003; Womack, 2015). The majority of these studies focusses on the aerosol or spore concentration in the atmosphere. However, also the emission patterns of the source organisms, i.e. the spore release characteristics depending on climatic conditions and the physiological activity status, might be relevant to explain the atmospheric concentrations. To fill that gap, a setup focused on the aerosol release patterns and mechanisms of the organisms could generate new insights into the relevance

of local bioaerosol emissions.

The Amazon Basin represents the World's largest rain forest with a dimension of about 6 000 000 km² (Melack and Hess, 2010). As tropical rain forests host an extensive spectrum of species and have an extremely high biomass turnover, bioaerosol transport from the biosphere into the atmosphere is high (Artaxo et al., 2013; Artaxo and Hansson, 1995; Crutzen et al., 1985; Simoneit et al., 1990). PBAP have been described to account for a major fraction of coarse mode aerosols in the Amazon

(Elbert et al., 2012; Moran-Zuloaga et al., 2018; Pöschl et al., 2010; Whitehead et al., 2016), and fungal spores have been suggested to be a key component in the coarse particle fraction (Elbert et al., 2007; Graham et al., 2003; Huffman et al., 2012a; Sesartic et al., 2013).

Cryptogams, including lichenized and non-lichenized fungi, bryophytes, and ferns cover the surfaces of up to 100 % of plants in tropical rain forests. They can be expected to play a major role in bioaerosol emissions, as they release reproduction

units as a result of sexual or asexual reproduction (Elbert et al., 2007; Fröhlich-Nowoisky et al., 2016; Fürnkranz et al., 2008; Richards, 1954). Cryptogamic organisms are mostly of poikilohydric nature, meaning that they are unable to actively regulate their water content, which causes their moisture content to strongly depend on the environmental humidity conditions. Accordingly, under dry conditions the organisms are metabolically inactive until they are reactivated by moisture uptake.

Fungi represent a separate kingdom, with taxa being present in all types of natural environments. Within the kingdom,

Ascomycota and Basidiomycota are the largest phyla (Mueller et al., 2004; Neves et al., 2013) besides several smaller ones. Microfungi, such as moulds, develop spores directly from hyphae, while other fungi colonize their substrate (e.g., dead wood or soil) by vegetative hyphae and form fruiting bodies of varying sizes for spore production and liberation. A single large fruiting body may release millions of spores in order to succeed in dispersal, multiplication, and reproduction (Webster and Weber, 2007). The reproduction units of fungi comprise spores resulting from sexual (e.g., basidiospores, ascospores) and

asexual modes of reproduction, such as teliospores, conidia, budding (yeast) cells, and fragments of hyphae (Cannon et al., 2018). Fungal spores in the atmosphere have been reported to reach number concentrations of $10^4$ to $10^5$ m$^{-3}$ in the Amazon (Elbert et al., 2007; Graham et al., 2003; Hoose et al., 2010). On a global scale, fungal spores may account for 8 to 186 Tg yr$^{-1}$ of bioaerosol emissions, which is roughly 23 % of total primary organic aerosol emission (Fröhlich-Nowoisky et al., 2016; Graham et al., 2003; Heald and Spracklen, 2009; Jacobson and Streets, 2009; Sesartic et al., 2013).

Species of Polyporales (Basidiomycota), the order of the model species of the present study, mostly colonize dead wood, where they form vegetative hyphae degrading the wood. On the surface of dead wood they form fruiting bodies with pores located at their lower side. The surface of these pores is covered by a hymenium, where basidiospores are produced on basidia. Each basidium externally forms mostly four asymmetrical basidiospores on little curved, tapering stalks, called sterigmata. The spores are fixed to the sterigmata at the hilum, which thus forms a characteristic feature of basidiospores.



In most terrestrial basidiomycetes the basidiospores are actively projected, thus being ballistospores, and several ejection mechanisms have been suggested (Webster and Weber, 2007). A widely accepted mechanism is that of surface tension catapult, originally described by Buller (1922) and Ingold (1939). For activation, a drop of water, the Buller's drop, forms at the hilar appendix and a second shallower liquid deposit forms on the surface of the spore above the hilar appendix (Fig. 1). Both drops

grow until they coalesce and then the basidiospore is immediately discharged (Pringle et al., 2005). This coalescence causes a sudden redistribution of mass and carries the spore and drop away from the sterigma. Experiments with the fungus *Itersonilia* sp. suggest that the liquid in both drops originates from water condensation upon the extrusion of a hygroscopic substance, e.g., mannitol and in some cases glucose. The spores measured were released from the sterigma at velocities above 1 m s$^{-1}$, but at low wind velocities there was a quick deceleration and loss of velocity (Webster and Weber, 2007). Basidiospore sizes range

between 3 and 20 µm (Webster and Weber, 2007). In accordance with Buller (Buller 1922, cited in Webster and Weber, 2007) the number of basidiospores produced by a single fruiting body can be extremely large, as, e.g., one cap of the mushroom *Agaricus campestris* has been calculated to produce ~ $2.6*10^9$ spores over the course of one day.

Water availability is a triggering factor for the formation of macroscopic fruiting bodies, where the spores are formed. Other parameters such as light, humidity, and temperature can influence spore liberation, and the simultaneous measurement of these

parameters under natural and controlled conditions is needed to elucidate the relevant interrelations (Neves et al., 2013).

The aim of this study was to establish measurement techniques, which allow a qualitative and quantitative assessment of spore emission patterns of non-lichenized fungi, lichens, bryophytes, and ferns under natural field conditions and under controlled environmental conditions in the lab. Measurements under field conditions allow following the spore release patterns over longer time spans during diel cycles and under changing environmental conditions, whereas lab measurements facilitate

a controlled analysis of the effect of particular environmental parameters and a detailed quantification of spore emissions. Direct observations of PBAP emission patterns will help to better understand spatiotemporal patterns in the abundance of the Amazonian coarse mode aerosol population.

## 2    Materials and methods

### 2.1    Study site

The experiments were conducted at the *Amazon Tall Tower Observatory* (ATTO) site in the central Amazon Basin, about 150 km northeast of the city of Manaus (Andreae et al., 2015). The trees of this plateau forest have an average height of 21 meters, grow at a density of up to 600 trees ha$^{-1}$, and with approx. 16 000 estimated tree species the Amazon is very diverse compared to other forest types (McWilliam et al., 1993; ter Steege et al., 2013). The precipitation has a seasonal minimum of around 47 mm per month in the dry season (August-November) and a maximum of 335 mm per month in the wet season

(February-May), according to measurements conducted from 1961 to 1990 (Andreae et al., 2015). Further relevant details on the research site can be found in previous publications (Andreae et al., 2015; Pöhlker et al., 2018).



## 2.2 Study organisms

The investigated model species was collected in the plateau forest area of the ATTO site during the dry season in August 2018. For the measurements under field and laboratory conditions, two specimens of the same species, growing in close vicinity to each other, were used for parallel measurements. According to the macro-morphological characteristics of the sampled specimen, the basidioma was classified as a polyporoid fungus (Fig. 2). Combined with micro-morphological details of the hymenium and basidiospores, the species was identified as *Rigidoporus microporus* (Polyporales, Basidiomycota), which has already been reported for the Amazon area (Gomes-Silva et al., 2014). In the results section, the exemplary measurements made with the model species are presented. Due to the exemplary nature of the data, replicate measurements have not been conducted here, but will be needed during an analysis of the emission patterns of different organisms.

## 2.3 Measurements under natural conditions

In a first approach, bioaerosol emission patterns were investigated with a mobile measurement setup directly at selected organisms in the field. The setup is illustrated in Fig. 3. For particle counting and sizing, an optical particle sizer (OPS model 3330, size range: 0.3 to 10 µm; TSI Inc., Shoreview, Minnesota, USA), was operated at a sampling interval of 30 seconds with an air flow of 1 L min$^{-1}$. This instrument classifies the detected particles based on aerosol light scattering into 16 size bins ranging from 0.3 to 10 µm. The OPS was characterized in more detail in Moran-Zuloaga et al. (2018). At the OPS inlet, an antistatic tube (inner diameter: 4 mm, length: ~1 m; conductive tubing, TSI Inc., Minnesota, USA) extended by a plastic funnel (diameter: 8 cm, Carl Roth GmbH Co, Karlsruhe, Germany) was installed. The funnel was placed at 2 to 5 cm distance from the investigated emission source (Fig. 3c). The measured particle concentrations were averaged over 5 minutes. For the fine and coarse mode, the particles in the size range from 0.3 to 1 µm and 1 to 10 µm, respectively, were summed and averaged over 5 minute intervals. The OPS instrument has a filter, where the sampled particles are collected. These filters cannot be directly used for microscopy, but the particles can be transferred to microscopy slides and subsequently analyzed by light microscopy and compared to imagery obtained from sections of the fungal hymenium, as shown in Figure 2d.

Air temperature and relative humidity (RH) were measured at 30-second intervals at a distance of approximately 5 to 10 cm from the emission source (Temperature/Relative Humidity Data Logger, HOBO U23 v2, Onset, Bourne, Massachusetts, USA; Fig. 3c). For weather protection, the instruments were installed in a metal housing (55 x 48 x 48 cm, Zarges GmbH, Weilheim, Germany; Fig. 3b). Thus, only the inlet and the sensor tip were placed outside the box, next to the investigated organism. The measurements were performed over 24 hours.

As accessory data on the mesoclimate, net radiation (in W m$^{-2}$) was measured at 75 m height (Net radiometer, NR-LITE2, Kipp & Zonen, Netherlands), precipitation (in mm min$^{-1}$) at 81 m height (Rain gauge TB4, Hydrological Services Pty. Ltd., Australia), and the occurrence of fog was detected with visibility measurements using an optical fog sensor installed at 43 m height (OFS, Eigenbrodt GmbH, Königsmoor, Germany). Fog events were defined to occur at visibility values below 2000 m. Data Loggers recorded the data at 1-minute intervals (CR3000 and CR1000, Campbell Scientific, Logan, Utah, USA; Andreae et al., 2015). Based on the field data 5-minute averages have been generated.



The time is presented in local time (LT), which is UTC-4 (UTC = Coordinated Universal Time).

## 2.4     Cuvette system for laboratory experiments

Measurements in the lab are used to verify the field data and to characterize the spore release patterns over a wider range of environmental conditions. For experiments under controlled laboratory conditions, we developed a cuvette system where the sample is placed in the air stream. As cuvette, a glass chamber was built from two horizontally arranged flat flange clear glass units (inner diameter 15 cm, total inner height 11 cm; Fig. 4), the upper one with three and the lower one with one opening. The openings of the upper lid were connected to an inlet tube providing outside air via a filter (pore diameter 0.3 µm, HEPA filter capsule, Pall, Dreieich, Germany) and to a sensor measuring RH and temperature inside the chamber (Datenlogger MSR 145, Seuzach, Switzerland). A third opening for further manipulations or measurements was closed with a glass stopper, as it was not in use. The opening of the lower lid was connected to an OPS (OPS 3330, TSI Inc. Shoreview, Minnesota, USA) via a conductive tube of approx. 10 cm length (inner diameter 4 mm, conductive tubing, TSI Inc., Minnesota, US). Inside the cuvette, the sample was placed on an aluminum grid, which allowed the emitted particles to drop into the outlet tube to be measured.

Measurements were performed over the course of 24 hours at a sampling interval of 10 seconds and an air flow of 1 L min$^{-1}$. Records obtained with an empty cuvette, which showed low particle numbers indicating an air-tight measurement setup and an undisturbed measurement signal, were used as blank/background signal. In a next step, a sample was placed inside the cuvette and investigated under natural moisture conditions, i.e. with the water content it had directly after field collection. In a third step, the sample was removed from the cuvette, sprayed with water until the outside of the sample was glossy wet and inserted in the cuvette again. In a final step the blank cuvette was measured again. Whereas steps 1 and 4 lasted for approx. 1 to 5 hours each, step 2 was run for ~ 4 and step 3 for ~ 17 hours. As the opening of the cuvette caused a short but strong increase of particles across all detected bins, these peaks were removed prior to further analysis of the data. In the presented example only the initial blank is shown, as the final blank showed identical low values, thus demonstrating reproducibility.

The water content (WC) of the samples was determined gravimetrically as percentage value based on the dry weight (% DW), determined after the measurements upon drying in an oven at 60°C until weight constancy. Also during lab measurements, the filter of the OPS could be used for microscopical analyses as described in section 2.3.

## 2.5     Particle collection and microscopic analysis

A custom-made impactor was used to collect samples of the released particles for later microscopic or chemical analysis. As impactor, a customized metallic capsule with plastic support for the sampling substrate was used, where round microscopy cover slides with a diameter of 12 mm were inserted as collection substrate. The impactor was connected to the laboratory experiment chamber via the tubing at the inlet of the lower lid, with a pump (M42x30/I, KAG, Hannover, Germany; flow rate: 8 L min$^{-1}$) following the impactor. Particle collection with the impactor was conducted for different periods (ranging between 5 s and 1 min) depending on the sample and its spore release behavior. For *Rigidoporus microporus*, the particle collection





was carried out for 30 s. From each investigated organism three slides were collected one after the other. The sample slides were stored in flat containers (Analyslide Petri Dishes, Polystyrol, 47 mm, Pall Life, Ann Arbor, Michigan, US) under dry conditions at room temperature (~ 25 °C) until further investigations.

For microscopic observation of the collected spores, each round cover slide was mounted onto a clear glass microscopy slide (26 x 76 mm, pre-cleaned, ready-to-use, Gerhard Menzel, Braunschweig, Germany). The optical analysis was carried out with a light microscope using magnifications of 400x and 1000x (Primo Star, Carl-Zeiss Microscopy, Jena, Germany). The images were captured with a coupled camera (Axiocam 105 color, Carl-Zeiss Microscopy, Jena, Germany) and processed using the software of the manufacturer (Zen software, 2.3 edition, Carl-Zeiss Microscopy, Jena, Germany).

## 2.6 Data and statistical analysis

The software IGOR Pro (Igor Pro 6.3.7, WaveMetrics. Inc, Lake Oswego, Oregon, USA) was used to calculate the 5-minute averages and help with the graphical analysis.

The software Statistica (13.3, StatSoft.Inc., Tulsa, Oklahoma, USA) was applied to test the normal distribution and the variability of the particle concentrations during the different steps of the laboratory experiment. A Chi-Square test was applied to test the normal distribution of the data and a Kruskal-Wallis test was used to test the difference between the blank cuvette, the naturally moist, and the artificially moistened sample. We applied a two-sided significance level of $p = 0.05$.

## 3 Initial results and discussion

### 3.1 Measurements under natural conditions

The following paragraph presents first results obtained by measurements of spore release patterns of one exemplary organism in the field, which illustrates the potential of this approach to explore the link between bioaerosol release and micrometeorological parameters in a systematic way.

During measurements of the model species under ambient conditions, neither fog nor rain occurred (Fig. 5). The coarse mode particles (Fig. 5b) exhibited the highest concentrations during nighttime, when the RH ranged between 73 and 96 %, with peak concentrations of $N_{1-10} = 600$ cm$^{-3}$, while the lowest numbers were measured during the day with only ~0.8 cm$^{-3}$ at RH values between 66 and 85 %. On average, coarse mode particle concentrations were $N_{1-10} = 330 \pm 105$ cm$^{-3}$ during nighttime (from 18:00 to 6:00 LT) and $N_{1-10} = 90 \pm 102$ cm$^{-3}$ during the day. Pronounced diurnal cycles of coarse mode particle abundance in the Amazon, such as the one found here (Fig. 5), have been described in previous studies (Huffman et al., 2012a; Moran-Zuloaga et al., 2018). These diurnal patterns might be driven by spore emission patterns, boundary layer dynamics (daytime dilution vs. nighttime concentration), or a combination of both. During convective daytime hours, the boundary layer top is located much higher than during night, which results in a dilution of the local emissions during the day and elevated concentrations during the night (Neves and Fisch, 2015). However, in our case the most important driver for the diurnal pattern of the coarse mode particles might be the release pattern of the fungus, as the measuring device was located in close vicinity





to the releasing fungus. Coarse mode particle occurrence in the open atmosphere is likely caused by a combination of particle release patterns and boundary layer height and effects.

In contrast to the coarse mode diurnal patterns, the fine mode concentrations were much lower with the highest fine mode concentration reached during the day ($N_{0.3-1} = 74$ cm$^{-3}$) and the lowest concentration during night ($N_{0.3-1} = 50$ cm$^{-3}$). On

average, the particle concentration in the fine mode reached $N_{0.3-1} = 64 \pm 9.2$ cm$^{-3}$ during the day and $N_{0.3-1} = 58 \pm 3.2$ cm$^{-3}$ during the night. Overall, the fine mode and the coarse mode particles show opposite diurnal trends, which indicates that both might be driven by different mechanisms. The fine mode particle concentration $N_{0.3-1}$ probably originated from biomass burning events, which occur at a higher frequency during the dry as compared to the wet season. The night-time minimum in $N_{0.3-1}$ can be explained by dry deposition of the fine mode aerosol to the forest canopy in the shallow nocturnal boundary layer.

The presented experimental approach of simultaneous observations of the particle size distribution, RH, temperature, and net radiation close to the organism allows to relate the organism's spore emission patterns to meteorological parameters. Based on a larger number of measurements from multiple organisms, which is subject of ongoing work, this is intended to provide a dataset of predictive value with regard to bioaerosol release and abundance. For further measurements, a combination of emission monitoring with a wider range of meteorological data should provide valuable insights into the detailed spore

emission patterns. As the geometric size of the spores of *Rigidoporus microporus* ranges between 3 and 5 µm (Gomes-Silva et al., 2014), they are expected to be measured in the coarse mode fraction. Upon an increase of RH in connection with a decrease of net radiation and temperature in the evening hours, the spore concentrations emitted by this sample increased immediately (Fig. 5). In contrast to that, spore concentrations decreased in the morning simultaneously to a decrease of the RH and an increase of the net radiation and temperature. Elevated basidiospore concentrations during nighttime hours have

previously been observed and were linked to the ballistic spore emission mechanism described above (Webster and Weber, 2007).

Rain and fog may also play an important role in modulating the bioaerosol release patterns (Huffman et al., 2013; Schumacher et al., 2013). Due to the absence of rain and fog during the time frame shown in Fig. 5, the relevance of these parameters on the spore release cannot be investigated in this particular case. Nevertheless, we believe that these parameters

could give valuable insights into particle release patterns and processes in other cases, which is also subject of ongoing work. The presented results have to be considered as exemplary and do not allow generalizations, as different emission patterns and relevant environmental parameters can be expected for different species, but they demonstrate the practicability of the measurement techniques.

## 3.2 Laboratory experiments

Measurements under laboratory conditions allowed a detailed investigation of the particle emission properties of a specific organism under controlled conditions, due to the exclusion of the surrounding emission sources. In combination with field measurements of the same specimen, the particle release process of the study organism can be separated from the environmental processes.





During the laboratory experiment, the initial blank showed average coarse and fine mode particle concentrations close to zero (both < 0.001 cm$^{-3}$, respectively; Fig. 6). In the coarse mode, the highest average particle concentrations were detected for the naturally moistened sample ($N_{1-10}$ = 10 cm$^{-3}$; p <0.001) and the artificially moistened sample ($N_{1-10}$ = 2.2 cm$^{-3}$), which were significantly higher than the blank ($N_{1-10}$ < 0.001 cm$^{-3}$; p<0.001). The coarse mode particle concentrations showed defined pulses when the sample was in the cuvette. Right after installation of the naturally moist sample in the cuvette, a defined peak spanning over ~2 hours occurred, reaching a maximum particle concentration of $N_{1-10}$ = 20 cm$^{-3}$. The WC of the sample ranged between 190 and 170 % and the RH ranged from 66 to 64 % during this time span. After installation of the artificially moistened sample there were two defined, but somewhat smaller pulses, still reaching maximum concentrations of $N_{1-10}$ = 13 and $N_{1-10}$ = 9.4 cm$^{-3}$ in the coarse mode, respectively. They had a duration of ~1.5 and ~2 hours, and during these times WC values ranged between 244-226 % and 197-169 %, and RH between 67-65 % and 63-62 %, respectively. The lowest particle concentration of $N_{1-10}$ <0.001 cm$^{-3}$, which is very close to zero, was measured from 6:00 to 11:00 a.m. while the WC ranged from 83 to 148 % and the RH between 52-59 %.

In the fine mode, particle concentrations throughout the experiment were very low (i.e., $N_{0.3-1}$ <0.001 cm$^{-3}$), which indicated the absence of leaks in the system and further suggests that no fine mode particles were co-emitted with the coarse mode pulses.

### 3.3 Particle collection

On the impactor samples, the particles were concentrated in one spot surrounding the impaction center. The amount of particles varied according to the sampling time and bioaerosol emission by the respective sample. Using the microscope at a magnification of at least 400x, it was possible to visualize the collected spores. For the examined species, we microscopically observed subglobose spores, about 3 x 5 µm in size, besides other random particles from the organism, e.g., fragments of hyphae about 1 to 8 µm in length, and undefined particles in smaller concentrations, ranging from 0.5 to 10 µm in size.

Impactor studies may also be useful during field measurements, as the coarse mode particles could be collected with the field setup. Another option is to use a new filter in the OPS and to investigate the spores collected on the filter by means of microscopy after sampling. In our field measurements, the contamination from surrounding emission sources was kept as small as possible by installing the inlet funnel of the detector close to the organism of interest. Collecting the released material with an impactor during the laboratory experiment minimizes contamination by other sources and allows a detailed observation of the particles (spores, cell fractions, etc) released by the investigated organism.

### 3.4 Overall discussion

The combined approach of field and the laboratory measurements seems to be very promising for investigations of the spore release patterns of spore producing organisms and their relevance for atmospheric spore concentrations. Whereas the field measurements are used to analyze the spore emission patterns of the investigated organisms under natural environmental conditions, the lab measurements facilitate analyses under controlled conditions over a wide range of environmental





characteristics. Thus, this combined approach allows a thorough characterization of the bioaerosol release patterns and mechanisms of defined organisms or communities. Up to now, bioaerosol measurement techniques were mainly used to measure atmospheric concentrations of overall aerosols in the atmosphere. Some of them distinguished between aerosols of biotic and abiotic origin, but we are not aware of techniques that were used to characterize the spore emission patterns of single

organisms. Thus, the strength of our approach is that it allows to characterize the spore release patterns of single organisms or cryptogamic communities under defined environmental conditions in the field and in the lab.

Measurements inside the cuvette allow close control of the environmental parameters, e.g. the air circulation, temperature, humidity, and light. As only particles in the coarse mode were found to be emitted, basidiospores seem to be the main emitted particle type, whereas fine mode particles seem to be of minor importance. The highest numbers of basidiospores were emitted

under naturally moist and artificially moistened conditions at water contents of the fungal fruiting body ranging between 244 and 169 %. Thus, the water content seems to play a relevant role in the spore emission process, which is in close agreement with other reports (Ehlert et al., 2017; Ingold, 1985; Sadyś et al., 2018) and with the fact that water condensation is essential for the active discharge of basidiospores. However, for this technique the fruiting bodies are mostly separated from their vegetative mycelium in the substrate (dead wood) that provides water, minerals, and nutrients for the fruiting body. This

separation may affect the release of spores or other particles. For the empty cuvette, concentrations of less than 0.2 cm$^{-3}$ particles were measured, indicating that the blanks serve as a good baseline for the laboratory measurements.

The measurements under field conditions showed a pronounced diurnal pattern in the coarse mode with the highest concentrations during nighttime, which has already been reported for other fungal species (Fernando et al., 2000; Gilbert and Reynolds, 2005; Stensvand et al., 1998). Aerosol and trace gas concentrations frequently have been shown to also display

diurnal and seasonal patterns (Gadoury et al., 1998; Huffman et al., 2012a; Moran-Zuloaga et al., 2018; Pöhlker et al., 2012; Yáñez-Serrano et al., 2015). However, in the natural environment there are many other emitting sources in the surrounding area, which might affect the measurement, and other environmental parameters, like the spectral composition of light (Ehlert et al., 2017; Igbalajobi et al., 2019; Pruß et al., 2014), might serve as particle release triggers, which have not been considered here. Other references already indicated RH or dew to be triggering factors for spore release (Stensvand et al., 1998). Overall,

the spore release patterns can be expected to show both interspecific and intraspecific variability. The monitoring of a larger variety of organisms might help to characterize the dominating release pattern of organisms in the environment and to identify the triggering factors.

In earlier studies, a vertical structure of aerosols within the canopy was shown in the tropics (Gilbert and Reynolds, 2005), which could be investigated by utilizing this method to monitor the organisms at varying canopy heights. If the results of spore

emission experiments are coupled with an inventory of cryptogams, very promising insights might be achieved. With both particle observation techniques, i.e., field and laboratory measurements, particles in the same size range have been sampled, confirming that the laboratory measurements can be considered as validation for the results obtained in the field. Additionally, the microscopic observation of the collected spores is fundamental for validation of the spore size and for identification of the organism being observed. Fine-mode particles were only sampled during the field experiment, indicating a different source for

this particle fraction. An experimental approach of screening different spore producing organisms at different height canopy

levels and during different seasons might promise new insights into the relevance of spore producing organisms for atmospheric aerosols and their effects on regional weather and climate.

**Data availability**

Shown data are provided in the supplement of this manuscript.

**Sample availability:**

Sample material of the investigated fungus can be loaned from the authors upon request.

**Supplement**

The data presented in the manuscript are given as 5-minute averages in the supplement.

**Team list**

Nina Löbs, Cybelli G. G. Barbosa, Sebastian Brill, David Walter, Florian Ditas, Marta de Oliveira Sá, Alessandro C. de Araújo, Leonardo R. de Oliveira, Ricardo H. M. Godoi, Stefan Wolff, Meike Piepenbring, Jürgen Kesselmeier, Paulo Artaxo, Meinrat O. Andreae, Ulrich Pöschl, Christopher Pöhlker, Bettina Weber

**Author contribution**

BW, NL, CP, SB, FD, and CGGB designed the experiments and SB, NL and CGGB carried them out. CGGB and MP
identified the fungus. MOS, ACA, and LRO measured, analyzed, and supplied the climate data. RHMG, SW, JK, PA, MOA, and UP gave valuable input regarding the design and realization of on-site bioaerosol measurements. NL and DW performed the formal analysis. NL, BW, and CP prepared the manuscript with contributions from all co-authors. NL, CGGB, and SB contributed equally to this manuscript.

**Competing interests**

The authors declare that they have no conflict of interest.

**Disclaimer**

The authors declare that they have no conflict of interest.



**Special issue statement**

This article is part of the special issue "Amazon Tall Tower Observatory (ATTO) Special Issue".

It is not affiliated with a conference.

**Acknowledgements**

We would like to acknowledge the German Federal Ministry of Education and Research (BMBF contracts 01LB1001A and 01LK1602B) and the Max Planck Society, supporting this project as well as the construction and operation of the ATTO site. We also acknowledge the support of the Brazilian Ministério da Ciência, Tecnologia e Inovação (MCTI/FINEP contract 01.11.01248.00) as well as the Amazon State University (UEA), FAPEAM, LBA/INPA and SDS/CEUC/RDS-Uatumã for their support during construction and operation of the ATTO site. We would like to thank Reiner Ditz, Susan Trumbore,

Alberto Quesada, Thomas Disper, and Hermes Braga Xavier for technical, logistical, and scientific support within the ATTO project, Thomas Klimach, Daniel Pickersgill, Isabella Hrabě de Angelis, Harald Paulsen, Peter Hoor, and Eckhard Thines for their technical and scientific expertise. NL would like to thank the Max Planck Graduate Center (MPGC) for the support. SB, CGGB, DW, FD, SW, UP, CP, and BW appreciate the support by the Max Planck Society. MdOS and LRdO would like to thank for the support of the Instituto Nacional de Pesquisas da Amazônia (INPA), RHMG would like to thank the Federal

University of Paraná, PA would like to thank funding from FAPESP (Fundação de Amparo à Pesquisa do Estado de São Paulo) through grant 2017/17047-0, and CNPq grant 425100/2016-2. ACdA would like to thank the Empresa Brasileira de Pesquisa Agropecuária (EMBRAPA), MP would like to thank the Goethe University Frankfurt/Main, MOA appreciates the support of the Max Planck Society and the University of San Diego, and BW would like to thank the Karl-Franzens-Universität Graz. This paper contains results of research conducted under the Technical/Scientific Cooperation Agreement between the National

Institute for Amazonian Research, the State University of Amazonas, and the Max Planck Society. The opinions expressed are the entire responsibility of the authors and not of the participating institutions.

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



**Figures**

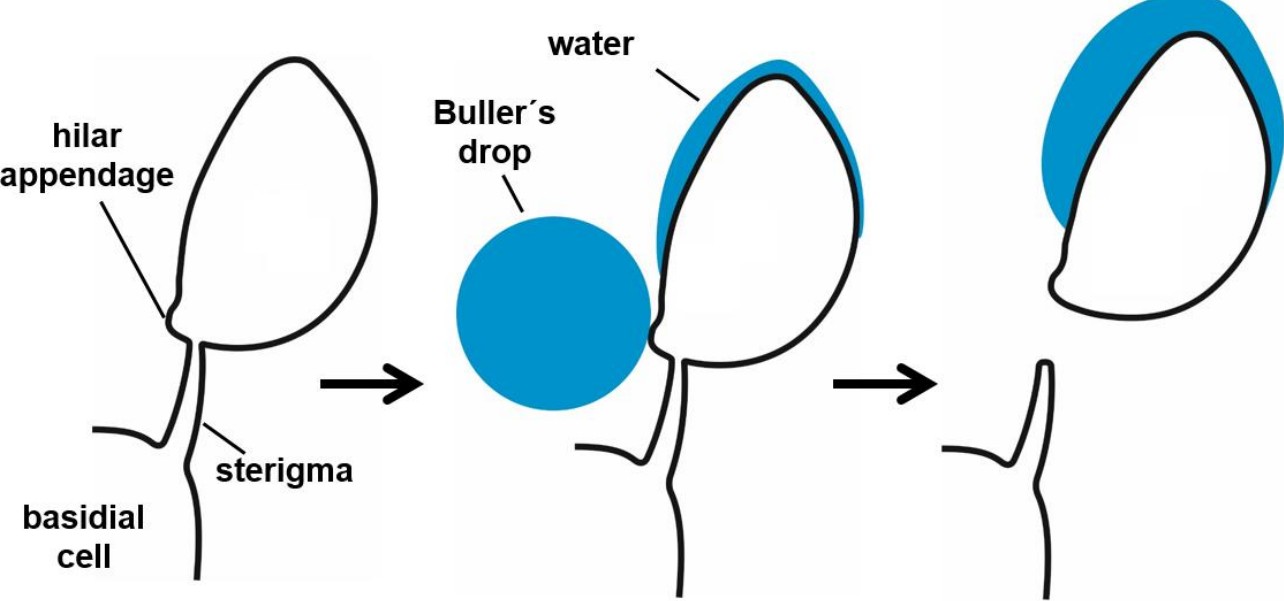

Fig. 1: Schematic drawing of the spore-building parts of basidiomycetous fungus. Active release mechanism based on a change of the gravity center due to the "Buller's drop" effect. Not to scale. Based on (Piepenbring, 2015).



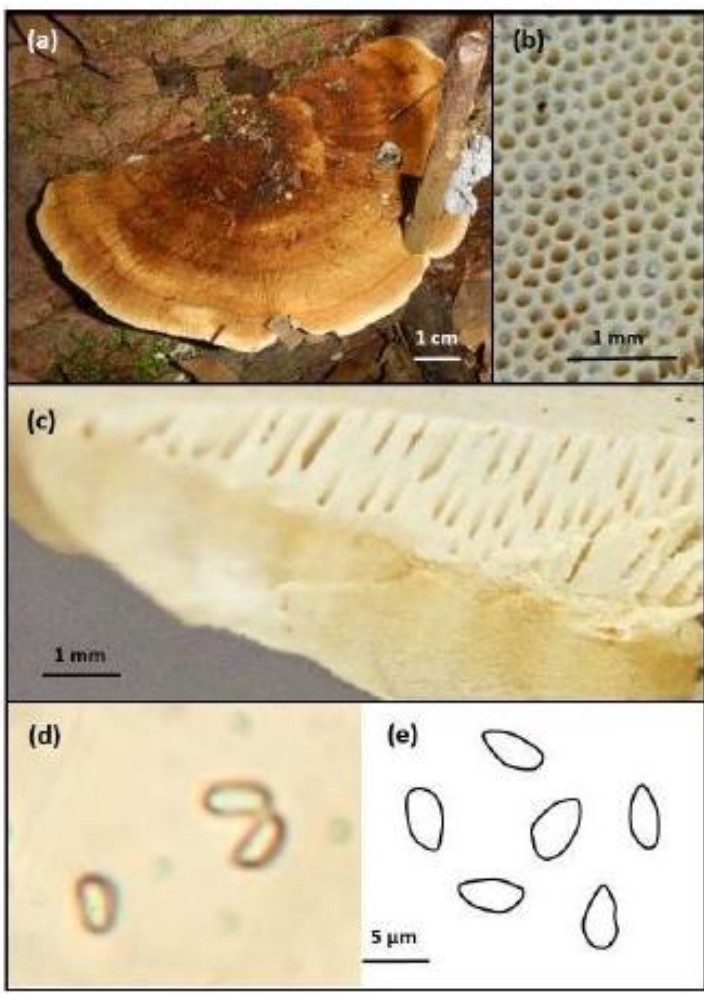

**Fig. 2:** Photographic documentation of the measured basidiomycete *Rigidoporus microporus* **(a)** Sessile basidioma growing on a dead hardwood, **(b)** polyporoid hymenium with white surface, **(c)** vertical cross-section of hymenium with homogeneous content, **(d)** microscopic image of the one-celled colorless spores at 1000 x magnification, **(e)** schematic drawing of some of the observed spores, average size 3 x 5 μm. Scales indicated.


**Figure 3: Setup used for field measurements. The optical particle sensor (OPS) is located in a box for weather protection. From this, an inlet ending with a funnel leads to the organism of interest. The setup is accompanied by a T/RH-sensor located in the direct vicinity of the sample (a) Schematic drawing, (b) overview photograph, and (c) close-up photograph of the described setup.**





**Figure 4: Setup used for laboratory experiments. The samples are placed in a glass cuvette in a vertical air stream. Air is supplied via an inlet with a filter connected to it; the outlet is connected to an OPS for particle measurements. A humidity and temperature sensor is placed inside the cuvette through an opening in the upper lid, which is sealed to avoid contamination. (a) Schematic drawing, and (b) close-up photograph.**

**Figure 5: Particle emission patterns of *Rigidoporus microporus* measured under field conditions over the course of 24 hours. (a) Temperature (*T*) and relative humidity (*RH*) measured next to the sample organism and net radiation (*R*ₙ) measured at 75 m height, (b) concentration of different particle size classes over time, and (c) number of particles ranging from 0.3 to 1 μm (fine mode) and**

5     **from 1 to 10 μm diameter (coarse mode). Data are presented as 5-minute averages.**



**Figure 6: Particle emission patterns of *Rigidoporus microporus* measured under lab conditions. (a) Temperature (*T*) and relative humidity (*RH*) measured inside the cuvette and gravimetrically determined water content of the sample (*WC*sample), (b) concentration of different particle size classes over time, and (c) number of the particles ranging from 0.3 to 1 μm (fine mode) and from 1 to 10 μm diameter (coarse mode). In the beginning and at the end of the measurement, a blank measurement was conducted with the empty cuvette (see labelling in upper part of figure; final blank not shown for brevity). After the initial blank, the sample was first measured with its natural water content (natural moisture) and then after spraying with water (artificial moisture). During opening of the cuvette for insertion and removal of the sample, particles from outside caused a short increase in particles (measurement artefact), which was removed. Data are presented as 5-minute averages.**