# Peer review of "Aerosol measurement methods to quantify spore emissions from fungi and cryptogamic covers in the Amazon"

_Atmospheric Measurement Techniques, 2019_

## Referee Comment (RC1) · Anonymous Referee #3 · 26 Aug 2019

The manuscript details two measurement techniques to sample fungal spores both in field and in the laboratory. The manuscript itself is well written and the topic is of high scientific relevance, considered the potential impact of primary biological particulate matter on climatic feedback cycles. The referee, however, finds some shortcomings in the methodology that keep the results of the manuscript at the level of speculation, at least for the field sampling part. For these reasons, the referee suggests that the manuscript is not published in the present form, but also strongly suggests a resubmission of the present work when these shortcomings are addressed due to the interest of the addressed topic and the potentialities shown by the experimental set-up.

Specific Comments:

Page 5, Lines 5-6: The measurements have been done only for a single specimen in the field and another one in the laboratory. The lack of replicates makes it hard to determine if the results from the measurement methodology are overall consistent.

Page 5, Lines 14-15: While heuristically speaking the referee agrees with the authors that at such a short distance the main contributions should come from the fungal spores, there's no confirmation of that. The OPS cannot discriminate between spores, other biological particles and inorganic aerosols and therefore it is hard to discriminate between the background that's not due to the fungus' spore discharge and the spore release itself. This could have been made more robust either, as the authors themselves acknowledge at page 9 lines 16-18, by sampling the OPS filter or by adding an impactor in cascade to the OPS and examining the impacted particles via microscopy. As a first approximation it could have been also enough to sample multiple fungi or characterize the background by measuring in a relative fungi-free area. As it is, with a single specimen measured in the field and no kind of downstream validation, the results are speculative.

Page 6, Paragraph 2.5: The authors' state that impaction sampling has been done "from each investigated organism". This means that it has been done both in the field and in the lab? How? By putting the specimen into the metallic capsule? If yes how was this performed in the field? The impactor studies, as the authors's state themselves, could have been used to give much more weight to the OPS measurements if they have been performed on the field specimen itself (see also previous comment).

Page 8-9, Paragraph 3.2 (and relative discussion): While the authors state throughout the paper about the importance of the combined laboratory/on-site approach, the results and discussion do not really highlight this linkage. How are the results of the laboratory experiments linked with the concentrations measured in the field? The lack of emissions in the fine mode and the linkage between spore release and moisture/water

content is consistent with known literature data, which confirms the validity of the laboratory set-up, but how does it relate with the findings in the field? The referee suggests to clarify this aspect in order to make more robust the usage of a parallel on-site/in-lab set-up.

---

## Referee Comment (RC2) · Anonymous Referee #4 · 5 Nov 2019

This manuscript presents experimental results from an interesting study examining the effect of environmental conditions on fungal spore release, using a novel approach based on both, controlled laboratory experiments and measurement of these bioaerosol species in the field under real-world conditions. Although the study addressed only one type of fungi, the findings of the study are of interest not only to the bioaerosol community but the atmospheric aerosol community at large, as the characteristics of biological aerosol components are still poorly constrained, partly because of the immense variety of species but mainly because of the challenges in obtaining representative measurement results. Therefore, this manuscript is highly relevant to the advancement of understanding bioaerosol sources and characteristics, and should

be published in AMT, upon consideration of a few comments listed below.

Specific comments:

1. Page 8, Lines 3-9: The authors state that the lower night-time levels of fine mode particles might be due to scavenging by the forest canopy, off-setting the typical increase in concentrations during boundary layer development. However, the authors also mention biomass burning as a potential source of the fine mode particles, which could be produced at a higher rate during night-time smoldering burns. It would be interesting to see information about the burning activities in the area that might have impacted the sampling site, which could be added here, if available.

2. Page 9, Lines 22-27: The authors mention detailed observation of the particles, including microscopic examination, but how about measurement of molecular markers, such as sugar alcohols (e.g., arabitol and mannitol), sterols (e.g., ergosterol), or others, as well as total protein – were any attempts made to do a more detailed chemical characterization of the observed particles?

3. Page 10, Lines 8-9: How does the fact that only coarse mode particles were emitted indicate that mainly basidiospores were released?

4. Page 10, Lines 21-24: Indeed, various environmental parameters may affect the release patterns of fungal spores, including solar radiation, although some studies (e.g., Liang et al., JAS, 66, 179–186, 2013) did not observe any relationship between ambient spore concentrations and solar radiation. As mentioned here, relative humidity is an important factor, specifically for fungal species utilizing wet spore discharge mechanisms. This has been seen in other ambient measurements as well, such as those reported by Gosselin et al. (ACP, 16, 15165–15184, 2016), Liang et al. (JAS, 66, 179–186, 2013), or Zhang et al. (ERL, 5, 024010, 2010).

---

## Author Comment (AC1) · 18 Nov 2019

This is the author's response to the comments of the Anonymous Reviewer #3; Submitted on 26 August 2019.

——————————————————

Referee comment: The manuscript details two measurement techniques to sample fungal spores both in field and in the laboratory. The manuscript itself is well written and the topic is of high scientific relevance, considered the potential impact of primary biological particulate matter on climatic feedback cycles. The referee, however, finds

some shortcomings in the methodology that keep the results of the manuscript at the level of speculation, at least for the field sampling part. For these reasons, the referee suggests that the manuscript is not published in the present form, but also strongly suggests a resubmission of the present work when these shortcomings are addressed due to the interest of the addressed topic and the potentialities shown by the experimental set-up.

Author response: We would like to thank the reviewer for his/her overall positive comments and the effort he/she spent on our manuscript.

———————————————————

Referee comment: Specific comments: Page 5, Lines 5-6: The measurements have been done only for a single specimen in the field and another one in the laboratory. The lack of replicates makes it hard to determine if the results from the measurement methodology are overall consistent.

Author response: As we have already explained during the initial manuscript evaluation, our manuscript is a technical proof of concept study written for the technical journal AMT. Purposefully, it was written without an extensive body of measurement data to focus on the actual technique. Replicates and extended discussion of the variability will be part of follow-up studies in the same context.

———————————————————

Referee comment: Page 5, Lines 14-15: While heuristically speaking the referee agrees with the authors that at such a short distance the main contributions should come from the fungal spores, there's no confirmation of that. The OPS cannot discriminate between spores, other biological particles and inorganic aerosols and therefore it is hard to discriminate between the background that's not due to the fungus' spore discharge and the spore release itself. This could have been made more robust either, as the authors themselves acknowledge at page 9 lines 16-18, by sampling the OPS filter

or by adding an impactor in cascade to the OPS and examining the impacted particles via microscopy. As a first approximation it could have been also enough to sample multiple fungi or characterize the background by measuring in a relative fungi-free area. As it is, with a single specimen measured in the field and no kind of downstream validation, the results are speculative.

Author response: In our statement to the comment during the initial manuscript evaluation we already explained that the focus of this study is to present the new measurements techniques. Whereas we believe that our exemplary measurements are correct, they should not be used for any further studies, as no replicate measurements nor statistical analyses have been made and they are only for one single fungal species. However, it indeed is important to make sure that the organism of interest is sampled with the OPS during field measurements. This can be verified by several different methods. First, one can make a section of the hymenium of the investigated fungus and by means of microscopy, the spores of the investigated fungus can be measured and characterized. This method was chosen in the current manuscript and a hymenium section and the spores investigated by microscopy are shown in Figure 2. The particles measured during field and lab measurements should be in the same size range as the fungal spores visualized under the microscope. Second, OPS filters newly installed before field or lab measurements can be used for microscopy to analyze them and to compare the collected particles with the spores observed under the microscope. Third, an impactor can be used during the lab measurements to collect an aerosol sample, which could be subsequently analyzed by means of microscopy. We consider that at least one of these alternative methods should be used to validate the measurements. We refer to these different possibilities in different parts of the manuscript: Description of methods "measurements under natural conditions" (page 5, line 20-22): "The OPS instrument has a filter, where the sampled particles are collected. These filters cannot be directly used for microscopy, but the particles can be transferred to microscopy slides and subsequently analyzed by light microscopy and compared to imagery obtained from sections of the fungal hymenium, as shown in Figure 2d." Description of

methods "Cuvette system for laboratory experiments" (page 6, line 24-25): "Also during lab measurements, the filter of the OPS could be used for microscopical analyses as described in section 2.3." The particle collection with an impactor is described in section 2.5 of the manuscript.

———————————————

Referee comment: Page 6, Paragraph 2.5: The authors' state that impaction sampling has been done "from each investigated organism". This means that it has been done both in the field and in the lab? How? By putting the specimen into the metallic capsule? If yes how was this performed in the field? The impactor studies, as the authors's state themselves, could have been used to give much more weight to the OPS measurements if they have been performed on the field specimen itself (see also previous comment).

Author response: As already explained by us during the initial manuscript evaluation, the impactor sampling can be used both in the field and the lab setup. In the manuscript, we describe it for the lab setup and here sampling was conducted for each organism. As described in the methods section (page 6, line 29-32), the sample is placed in the chamber and the impactor is installed at the inlet of the lower lid with a pump following the impactor. As stated above and explained in the manuscript, there are various methods of sampling validation. For our exemplary sampling we made a hymenium section and investigated the fungus by means of microscopy (see Fig. 2). The fungal spores had a diameter of ∼3-5 $\mu$m, which was also the diameter of the spores collected during the laboratory and field measurements, and the particle concentrations were clearly constrained to that narrow particle window, thus verifying the investigated fungus as particle source.

———————————————

Referee comment: Page 8-9, Paragraph 3.2 (and relative discussion): While the authors state throughout the paper about the importance of the combined laboratory/on-

site approach, the results and discussion do not really highlight this linkage. How are the results of the laboratory experiments linked with the concentrations measured in the field? The lack of emissions in the fine mode and the linkage between spore release and moisture/water content is consistent with known literature data, which confirms the validity of the laboratory set-up, but how does it relate with the findings in the field? The referee suggests to clarify this aspect in order to make more robust the usage of a parallel on-site/in-lab set-up.

Author response: This comment has also been addressed by us during the initial manuscript evaluation. We made the connection between the field and lab measurements much clearer in the methods section. In addition, we also clarified that in the overall discussion (page 9, line 30 – page 10, line 2): "Whereas the field measurements are used to analyze the spore emission patterns of the investigated organisms under natural environmental conditions, the lab measurements facilitate analyses under controlled conditions over a wide range of environmental characteristics. Thus, this combined approach allows a thorough characterization of the bioaerosol release patterns and mechanisms of defined organisms or communities." Also later in the overall discussion, this issue is clearly stated (page 10, line 30-35): "With both particle observation techniques, i.e., field and laboratory measurements, particles in the same size range have been sampled, confirming that the laboratory measurements can be considered as validation for the results obtained in the field. Additionally, the microscopic observation of the collected spores is fundamental for validation of the spore size and for the identification of the organism being observed. Fine-mode particles were only sampled during the field experiment, indicating a different source for this particle fraction."

---

## Author Comment (AC2) · 18 Nov 2019

* * *
This is the author's response to the comments of the Anonymous Reviewer #4; Submitted on 04 November 2019.
* * *
Referee comment: This manuscript presents experimental results from an interesting study examining the effect of environmental conditions on fungal spore release, using a novel approach based on both, controlled laboratory experiments and measurement of these bioaerosol species in the field under real-world conditions. Although the study

addressed only one type of fungi, the findings of the study are of interest not only to the bioaerosol community but the atmospheric aerosol community at large, as the characteristics of biological aerosol components are still poorly constrained, partly because of the immense variety of species but mainly because of the challenges in obtaining representative measurement results. Therefore, this manuscript is highly relevant to the advancement of understanding bioaerosol sources and characteristics, and should be published in AMT, upon consideration of a few comments listed below.

Author response: We would like to thank reviewer #4 for his/her positive evaluation of our manuscript and his/her very helpful suggestions.

————————————————

Referee comment: Specific comments: 1. Page 8, Lines 3-9: The authors state that the lower night-time levels of fine mode particles might be due to scavenging by the forest canopy, off-setting the typical increase in concentrations during boundary layer development. However, the authors also mention biomass burning as a potential source of the fine mode particles, which could be produced at a higher rate during night-time smoldering burns. It would be interesting to see information about the burning activities in the area that might have impacted the sampling site, which could be added here, if available.

Author response: Measurements with a Multi Angle Absorption Photometer (MAAP; Thermo Scientific) revealed that during the reported period the concentrations of black carbon indeed were rather high, with values ranging between 0.9-1.3 $\mu$g m-3. We now included this information in the manuscript in the following way (page 8. Line 10-13): "The fine mode particle concentration N0.3-1 probably originated from biomass burning events, as simultaneous online measurements with an MAAP instrument (Thermo Scientific, MA, USA) revealed elevated values of black carbon ranging between 0.9 – 1.3 $\mu$g m-3. Biomass burning events generally occur at a higher frequency during the dry as compared to the wet season."

————————————————

Referee comment: 2. Page 9, Lines 22-27: The authors mention detailed observation of the particles, including microscopic examination, but how about measurement of molecular markers, such as sugar alcohols (e.g., arabitol and mannitol), sterols (e.g., ergosterol), or others, as well as total protein – were any attempts made to do a more detailed chemical characterization of the observed particles?

Author response: This indeed is an excellent suggestion, which could be followed during later research. We added this idea to the discussion section (page 9, line 32 – page 10, line 2): "Additionally, the collected material can be used for a chemical characterization of the observed particles by analyzing molecular markers, like sugar alcohols (e.g., arabitol, mannitol), sterols (e.g., ergosterol) and others, as well as an assessment of the total protein content."

————————————————

Referee comment: 3. Page 10, Lines 8-9: How does the fact that only coarse mode particles were emitted indicate that mainly basidiospores were released?

Author response: This sentence indeed might be somewhat unclear and thus was reformulated by us in the following way (page 10, line 13-15: "The measured particles covered a narrow size range (2.5 – 4 $\mu$m diameter) in the coarse mode, which conforms to the spore size of the investigated fungus, whereas fine mode particles seemed to be of minor importance."

————————————————

Referee comment: 4. Page 10, Lines 21-24: Indeed, various environmental parameters may affect the release patterns of fungal spores, including solar radiation, although some studies (e.g., Liang et al., JAS, 66, 179–186, 2013) did not observe any relationship between ambient spore concentrations and solar radiation. As mentioned here, relative humidity is an important factor, specifically for fungal species utilizing wet spore

discharge mechanisms. This has been seen in other ambient measurements as well, such as those reported by Gosselin et al. (ACP, 16, 15165–15184, 2016), Liang et al. (JAS, 66, 179–186, 2013), or Zhang et al. (ERL, 5, 024010, 2010).

Author response: Thank you very much for pointing out this interesting literature. We now cite it as additional literature suggesting relative humidity as an important factor.
* * *